# Tactile MNIST:
# Benchmarking Active Tactile Perception

**Tim Schneider**[1,2], **Guillaume Duret**[2], **Cristiana de Farias**[1],
**Roberto Calandra**[3], **Liming Chen**[2], **and Jan Peters**[4]

[1]Department of Computer Science, TU Darmstadt, Germany.
[2]LIRIS, CNRS UMR5205, Ecole Centrale de Lyon, France.
[3]LASR Lab & CeTI, TU Dresden, Germany.
[4]DFKI, Hessian.AI, and Centre for Cognitive Science, TU Darmstadt, Germany.

## Abstract

Tactile perception has the potential to significantly enhance dexterous robotic manipulation by providing rich local information that can complement or substitute for other sensory modalities such as vision. However, because tactile sensing is inherently local, it is not well-suited for tasks that require broad spatial awareness or global scene understanding on its own. A human-inspired strategy to address this issue is to consider active perception techniques instead. That is, to actively guide sensors toward regions with more informative or significant features and integrate such information over time in order to understand a scene or complete a task. Both active perception and different methods for tactile sensing have received significant attention recently. Yet, despite advancements, both fields lack standardized benchmarks. To bridge this gap, we introduce the *Tactile MNIST Benchmark Suite*, an open-source, Gymnasium-compatible benchmark specifically designed for active tactile perception tasks, including localization, classification, and volume estimation. Our benchmark suite offers diverse simulation scenarios, from simple toy environments all the way to complex tactile perception tasks using vision-based tactile sensors. Furthermore, we also offer a comprehensive dataset comprising 13,500 synthetic 3D MNIST digit models and 153,600 real-world tactile samples collected from 600 3D printed digits. Using this dataset, we train a CycleGAN for realistic tactile simulation rendering. By providing standardized protocols and reproducible evaluation frameworks, our benchmark suite facilitates systematic progress in the fields of tactile sensing and active perception.
**Project page:** https://sites.google.com/robot-learning.de/tactile-mnist

## 1 Introduction

Tactile perception is fundamental for enabling agents to interact effectively with their environments. Studies of humans with impaired touch reveal that they face significant challenges in grasping and performing routine manipulation tasks due to insufficient feedback about contact states between fingers and objects [4]. Moreover, touch often complements—or even substitutes—other sensory modalities such as vision: we feel the shape of a hard-to-see object on a high shelf, count cookies in a jar without looking, or locate a key inside a bag purely by touch. Unlike vision, which typically offers a broad field of view, touch provides highly localized yet information-rich feedback confined to the point of contact [5, 6]. It is this inherently interactive nature that enables agents (such as robots) to explore visually occluded areas, classify textures, infer material properties like stiffness

---

Corresponding Author: `tim@robot-learning.de`

Submitted to 39th Conference on Neural Information Processing Systems (NeurIPS 2025). Do not distribute.

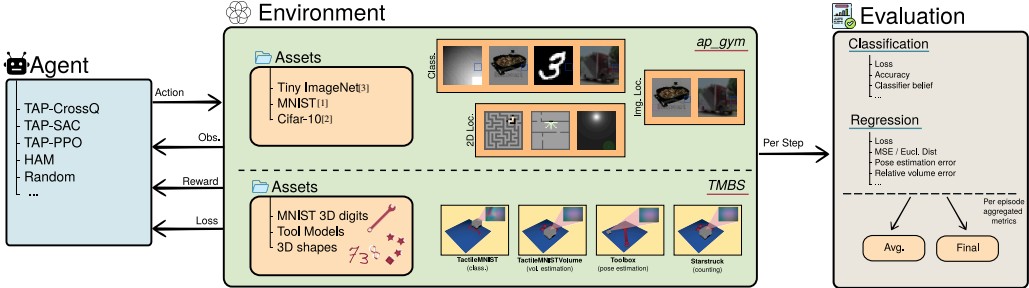

Figure 1: Overview of the *Active Perception Gym* (`ap_gym`), the *Tactile MNIST Benchmark Suite* (TMBS), and their associated assets. In the center we depict each environment from `ap_gym` and TMBS, along with the included asset sets (both custom and external). On the left, an agent (e.g., an active perception algorithm) interacts with the environment by receiving observations, rewards, and losses, and returning actions. On the right, we show task-specific evaluation metrics, available at each step, with support for both per-step outputs and aggregated performance scores.

and friction, and detect fine local features of objects [7, 8, 9, 10, 11, 12, 13, 14]. However, without standardized benchmarks and widely adopted datasets, it becomes difficult to rigorously evaluate new algorithms, reproduce results, or compare different approaches in a fair way. Yet, despite its growing importance, the field of tactile sensing still lacks such structured benchmarks and community-wide datasets tailored to touch-related tasks

In contrast to tactile sensing, computer vision has significantly benefited from such standardized datasets and clearly defined evaluation protocols. MNIST [1] established an early baseline for digit recognition that is still relevant today. Datasets such as ImageNet [15] and COCO [16] expanded both scale and complexity, driving major advances in object classification, detection, and scene understanding. Additionally, domain-specific benchmarks such as KITTI [17] and Omni3D [18] have enabled focused progress on challenges like fine-grained categorization and robustness in real-world applications. By comparison, the few tactile perception benchmarks in the literature still rely on custom hardware or narrowly scoped datasets, which limits their generalizability and slows broader adoption [14, 19].

In this work, we focus explicitly on benchmarking *active tactile perception* tasks where agents deliberately interact with their environment to gather task-relevant information. Active perception involves strategic decisions about where and when to sense, efficiently choosing actions that maximize information gain [10, 9, 20, 8]. Here, as each contact takes time, resources, and may cause wear on sensors or environments, efficiency becomes a central concern: agents must extract as much information as possible using as few interactions as necessary. A well-designed benchmark for active tactile perception can help answer key questions such as: How should an agent select contact points to maximize information gain? What policies enable accurate inference with minimal touch? How does uncertainty (from sensor noise, ambiguous contact, or environmental variability) affect the trade-off between exploration and confidence?

We introduce *Active Perception Gym* (`ap_gym`)[1], a framework compatible with Gymnasium [21], designed to benchmark active perception algorithms. `ap_gym` includes nine toy scenarios where an agent must learn to efficiently extract information to solve perception tasks. Building on `ap_gym`, we present the *Tactile MNIST Benchmark Suite* (TMBS)[2], which extends the framework to simulated active tactile perception problems. In TMBS, agents control a simulated GelSight Mini sensor [22] without access to visual inputs. Tactile perception tasks include MNIST-style classification of 3D digit models, pose estimation of tools on platforms, and object counting. Across all tasks, the agent must actively determine *what* to sense and strategically select *where and when* to explore through touch. Thus, solving these tasks requires solving a dual problem: making accurate predictions from past observations while optimizing an exploration policy to maximize information gain. An overall visualization of our framework and tasks in `ap_gym` and TMBS is shown in Fig. 1.

---

[1] https://github.com/TimSchneider42/active-perception-gym
[2] https://github.com/TimSchneider42/tactile-mnist

Table 1: Comparison of Active Tactile Perception Benchmarks. Tactile modalities are **bolded**.

| Method | Dataset Available | Active Benchmark | Sensor Modality |
|---|---|---|---|
| Active Vision Grasp [24] | ✗ | ✓ | Vis |
| R3ED [25] | ✓ | ✓ | Vis |
| Robotic Vision Challenge [26] | ✗ | ✓ | Vis |
| NBV_Bench [27] | ✓ | ✓ | Vis |
| AVD [28, 29] | ✓ | ✓ | Vis |
| Active Object Search [30] | ✓ | ✓ | Vis |
| ActiView [31] | ✓ | ✓ | Vis+Text |
| TIP Bench. [32] | ✗ | ✗ | **Tac** |
| FoTa [33] | ✓ | ✗ | **Tac** |
| YCB-Slide [34] | ✓ | ✗ | **Tac** |
| TacBench [14] | ✓ | ✗ | **Tac** |
| ActiveCloth [35] | ✓ | ✗ | **Tac** |
| Touch and Go [36] | ✓ | ✗ | **Tac**+Vis |
| FeelSight [19] | ✓ | ✗ | **Tac**+Vis |
| ViTac [37] | ✗ | ✗ | **Tac**+Vis |
| SSVTP [13] | ✓ | ✗ | **Tac**+Vis |
| GelFabric [38] | ✓ | ✗ | **Tac**+Vis |
| VisGel [39] | ✓ | ✗ | **Tac**+Vis |
| PHYSICLEAR [40] | ✓ | ✗ | **Tac**+Text |
| TVL [41] | ✓ | ✗ | **Tac**+Vis+Text |
| Touch100k [42] | ✓ | ✗ | **Tac**+Vis+Text |
| ObjectFolder [43, 44, 45] | ✓ | ✗ | **Tac**+Vis+Audio |
| **Ours** | ✓ | ✓ | **Tac** |

With this benchmark, our goal is to provide a reproducible and accessible evaluation framework for the tactile perception community. To that end, TMBS is a fully *simulated* environment that is easy to set up and enables rapid, consistent comparisons, without the need to replicate a complex real-world setup. However, we acknowledge the inherent challenges and noise present in real-world tactile data that are often absent in simulation. To help bridge this sim-to-real gap, we complement our simulated environment with a large-scale dataset of 13,580 high-fidelity 3D object models. From this collection, we 3D-printed 600 objects and constructed a curated real-world dataset comprising 153,600 tactile contacts, each annotated with detailed temporal and spatial metadata. We further leverage this dataset to train a CycleGAN [23], enabling the rendering of realistic tactile signals within the simulation.

In summary, our main contributions are:

- To the best of our knowledge, we introduce the first benchmark suite specifically for active *tactile* perception. It offers a range of tasks from simple toy problems to challenging, high-dimensional scenarios.

- We further introduce `ap_gym`, an extensible framework for generic active perception algorithms. `ap_gym` is Gymnasium compatible and is, thus, easy to integrate with existing reinforcement-learning pipelines, enabling fair evaluation.

- We provide an open dataset of 13,580 high-resolution 3D models of handwritten digits, designed for both simulated tactile-image generation and physical 3D printing.

- From our library of 3D models, we 3D-printed 600 objects and captured 153,600 tactile contacts using a GelSight Mini sensor, each annotated with spatial location and class labels.

## 2 Related Work

**(Active) Tactile Perception:** Tactile sensing enables robots to infer object geometry, texture and material properties through physical contact, complementing or, in some cases, substituting vision. Vision-based tactile sensors such as GelSight [22] and DIGIT [46] have become widely available, producing high-resolution "tactile images" that capture local surface features and force distributions. These rich signals have been exploited for shape reconstruction and material recognition [7, 32, 41, 40], as well as advanced dexterous manipulation [47, 48, 19].

Much of the work in tactile sensing, however, focuses on passive touch: the robot either registers a single contact or follows a predefined exploration policy. Inspired by the successes of active vision (as well as by early research on active touch in robotics [49, 50]), recent efforts have revisited the idea of tactile exploration as an active, decision-driven process. In this framing, the robot dynamically selects where and how to touch next, rather than relying on a fixed sequence of actions. Gaussian

process and Bayesian optimization have been employed to drive this active exploration, yielding significant improvements in tasks such as shape reconstruction, texture classification and grasp planning [10, 7, 51]. Reinforcement-learning based approaches [9] tackle active exploration in high-dimensional tactile state spaces. Furthermore, [8] introduced HAM a selective-attention mechanism to optimize scene exploration, and in [52] task-agnostic strategies generalize active touch across different objectives. Additionally, [35] demonstrated how Kinect-based vision can guide active touch for material classification, and [13] proposed a self-supervised visuo-tactile pretraining scheme that benefits both passive and active perception tasks.

**Benchmarking Methods for Tactile Sensing & Active Perception:** Over the past decade, the active vision community has produced a number of datasets and challenges to evaluate a range of tasks. Early work such as the Active Vision Dataset (AVD) provided large-scale Kinect captures for navigation and class-incremental learning tasks [28, 29]. Subsequent efforts explored next-best-view planning for classification (NBV_Bench [27]), heuristic and data-driven view selection for grasp synthesis on YCB objects [24], and embodied 3D exploration in real indoor scenes (R3ED [25]). Simulation-based approaches such as the Robotic Vision Scene Understanding Challenge [26] and Active Object Search [30] have further expanded evaluation protocols for semantic SLAM and object detection tasks. More recently, multi-modal active perception has been addressed by ActiView, which tests an agent's ability to zoom and pan to answer vision-language queries [31].

In parallel, tactile sensing research has released a diverse set of characterization benchmarks (TIP Bench. [32]), texture and material recognition datasets (ActiveCloth [35], ViTac [37], SSVTP [13], GelFabric [38]), and cross-modal vision–touch benchmaks and datasets (VisGel [39], Touch and Go [36], PHYSICLEAR [40], FeelSight [19], TacBench [14]). Large-scale multimodal datasets such as Touch100k [42], TVL [41], and FoTa [33] now exceed tens of thousands of samples across vision, touch, and language modalities. Multisensory datasets like ObjectFolder [43, 44, 53] further integrate tactile, visual, and audio data. In a similar manner, efforts to standardize vision-based tactile simulation have led to platforms like TACTO [54] and Taxim [55], which enable high-resolution visuo-tactile data generation. Despite these efforts, only the datasets provided by ActiveCloth [35] and SSVTP [13] support downstream active tactile perception tasks. However, these works do not provide standardized evaluation protocols or benchmarks to systematically evaluate active perception approaches. A comparison of existing datasets and benchmarks is summarized in Table 1. To the best of our knowledge, our proposed Tactile MNIST Benchmark Suite is the first to introduce a dedicated and reproducible benchmark for *active tactile perception*, where tactile exploration is an integral component of the perceptual process.

Table 2: Overview of the environments, including task types, descriptions, and assets.

| Suite | Environment | Task Type | Description | Assets |
|---|---|---|---|---|
| ap_gym | TinyImageNet | Classification | Classify natural images into 200 categories by moving a limited field-of-view glimpse. | Tiny ImageNet [3] |
| | CIFAR10 | Classification | Classify natural images into 10 categories by moving a limited field-of-view glimpse. | CIFAR-10 [2] |
| | CircleSquare | Classification | Determine whether a given image contains a circle or a square using limited agent visibility. | Geometric shapes |
| | MNIST | Classification | Digit recognition task using standard MNIST digits. | MNIST [1] |
| | LightDark | Regression (2D localization) | Position estimation from brightness-dependent noisy observations, requiring movement to light. | None |
| | LIDARLocRooms | Regression (2D localization) | Navigate procedurally generated maps with ambiguous LIDAR readings to localize. | None |
| | LIDARLocMaze | Regression (2D localization) | Navigate procedurally generated mazes with ambiguous LIDAR readings to localize. | None |
| | TinyImageNetLoc | Regression (2D patch localization) | Localize a glimpse within a natural image by moving a limited field-of-view. | Tiny ImageNet [3] |
| | CIFAR10Loc | Regression (2D patch localization) | Localize a glimpse within a natural image by moving a limited field-of-view. | CIFAR-10 [2] |
| TMBS | TactileMNIST | Classification | Touch-based digit classification using a vision-based tactile sensor. | MNIST 3D digits |
| | TactileMNISTVolume | Regression (volume estimation) | Estimate volume of digits using a vision-based tactile sensor. | MNIST 3D digits |
| | Toolbox | Regression (object pose estimation) | Estimate 6D pose of tools (e.g., a wrench) using a vision-based tactile sensor. | 3D tools |
| | Starstruck | Classification (counting) | Count stars among other objects using a vision-based tactile sensor. | 3D shapes |

## 3 Framework: Benchmarking Active Perception

In this section, we introduce both the *Active Perception Gym* (`ap_gym`), a framework for benchmarking active perception algorithms, and the *Tactile MNIST Benchmark Suite* (TMBS), which introduces environments for four active tactile perception tasks. Fig. 1 depicts an overview of our framework.

### 3.1 Active Perception

In active perception tasks, an agent's main objective is to gather information and make predictions about a desired property of the environment, e.g., the class label or pose of an object. Examples of such properties could be the location of an object in case of a search task or the class of an object for the agent in case of a classification task. To gather information, the agent must interact with the environment, e.g., by moving a sensor around a platform in case of TMBS.

We model active perception as an episodic process, where the agent can take a number of time-discrete sequential actions until the episode terminates and is reset. At every step, the agent obtains an observation (e.g., a tactile glimpse) from the environment that reveals some information but never the full state at once. Formally, that makes active perception problems a special case of Partially Observable Markov Decision Processes (POMDPs).

POMDPs are defined by the tuple $(S, A, T, R, \Omega, O, \gamma)$, consisting hidden states $S$, actions $A$, a transition function $T : S \times A \times S \to [0, 1]$, a reward function $R : S \times A \to \mathbb{R}$, a set of observations $\Omega$, an observation function $O : S \times A \times \Omega \to [0, 1]$, and a discount factor $\gamma \in [0, 1]$. The objective of the agent in a POMDP is to maximize the expected cumulative reward over time by selecting actions based on its belief about the underlying state. Since the agent does not have direct access to the true state, it maintains a belief distribution over states, updating it using observations and the observation function. The environment evolves according to the transition function, where taking an action leads to a probabilistic transition to a new state, which in turn generates an observation based on the observation function.

In case of active perception problems, we assume that the hidden state $S$, the action $A$, the reward function $R$, and the transition function $T$ have specific structures. First, we assume that the target property the agent is tasked to predict is part of the hidden state. Hence, $S$ is defined as $S = S_{\text{base}} \times Y^*$, where $S_{\text{base}}$ is the set of base (hidden) states of the environment and $Y^*$ is the set of prediction targets. E.g., $Y^*$ could be the set of classes in a classification task or the set of possible locations in a localization task, while $S_{\text{base}}$ contains all the other hidden state information. To allow the agent to make predictions, the action space $A$ is defined as $A_{\text{base}} \times Y$, where $A_{\text{base}}$ is the base action space and $Y$ is the prediction space. The base action space $A_{\text{base}}$ contains all the actions the agent can take to interact with the environment, while $Y$ is the set of possible predictions the agent can make. Crucially, environments are defined in a way that the agent's prediction never influences the hidden state of the environment. Thus, the transition function $T$ is defined as $T(s, a, s') = T(s, (a_{\text{base}}, y), s') = T_{\text{base}}(s, a_{\text{base}}, s')$. An example of a base action could be a desired movement of the tactile sensor, while the prediction could be the logits of the agent's current class prediction.

Finally, the reward function is defined as $R(s, a) = R((s_{\text{base}}, y^*), (a_{\text{base}}, y)) = R_{\text{base}}(s_{\text{base}}, a_{\text{base}}) - \ell(y^*, y)$, where $R_{\text{base}}$ is the base reward function and $\ell$ is a differentiable loss function. An example

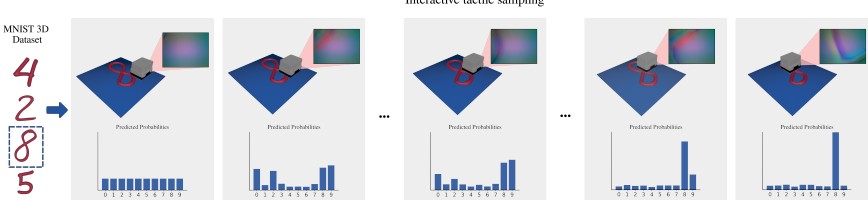

Figure 2: Illustration of the TactileMNIST classification task. In each episode of the TactileMNIST classification task, one random digit from the MNIST 3D dataset is selected and presented to the agent. It can then move the sensor around and touch the object 16 times before the episode terminates. Notably, it does not receive any visual input but has to rely solely on the readings from its tactile sensor. After every touch, it has to make a prediction about the class label, the digit's numeric value, and its performance is measured by the average prediction accuracy throughout the episode.

173 for a base reward could be an action regularization term, while the loss function $\ell$ could be a
174 cross-entropy loss in a classification task. Hence, the agent has to make a prediction in every
175 step, encouraging it to gather information quickly to maximize its prediction reward early on. A
176 visualization of this process on the TactileMNIST digit classification task is shown in Fig. 2.

## 3.2 Active Perception Gym

177

178 `ap_gym` models active perception tasks as episodic processes in a way that is fully compatible with
179 Gymnasium [21]. Each task is defined as a Gymnasium environment, bundled with the differentiable
180 loss function $\ell(y^*, y)$ and the prediction target $y^*$. Since the loss functions need to convey gradient
181 information to the learning algorithm, we currently provide them either JAX [56] or PyTorch [57]
182 functions, but more autograd frameworks might be supported in the future. During roll-outs, `ap_gym`
183 automatically computes task-dependent metrics, such as accuracy for classification, or Euclidean
184 distance for regression.

185 `ap_gym` provides a family of lightweight environments designed to isolate core exploration and
186 decision-making behaviors in active perception. As summarized in Table 2, `ap_gym` includes 11
187 environments spanning both classification and regression tasks. Four progressively harder image-
188 based classification benchmarks (CircleSquare, MNIST, CIFAR-10, TinyImageNet) evaluate an
189 agent's ability to select informative glimpses from natural or synthetic visuals. Two image-localization
190 tasks (TinyImageNetLoc, CIFAR10Loc) require the agent to infer the position of a limited-field-
191 of-view patch within a larger image. Finally, five non-visual regression tasks — LightDark and
192 four LIDAR-based 2D localization environments (Rooms and Maze) — challenge agents to reduce
193 state uncertainty by navigating procedurally generated maps or lighting fields. Crucially, in the
194 self-localization environments, the agent influences the property it is trying to infer — its position —
195 through its actions. Hence, the prediction target changes over time in these environments, which is an
196 explicitly supported aspect of `ap_gym` environments.

197 For the image-based classification and localization tasks, `ap_gym` relies on a mix of third-party assets:
198 Tiny ImageNet [3], CIFAR-10 [2], and MNIST [1] datasets. The LIDARLoc (rooms and maze)
199 environments generate map layouts procedurally and require no external data. Whenever applicable,
200 `ap_gym` defines two versions of each environment, one for training with the training split of the
201 respective dataset, and one for evaluation with the test split.

202 By abstracting away complex contact models and dynamics, `ap_gym` environments enable rapid
203 prototyping of active perception strategies and provide a controlled baseline for more complex
204 scenarios in the TMBS suite. However, despite their simplistic appearance, all `ap_gym` environments
205 impose significant challenges due to partial observability and non-immediate action payoffs. The tasks
206 differ substantially from each other, testing the algorithm's capability to handle diverse scenarios.

## 3.3 The Tactile MNIST Benchmark Suite

207

208 Tactile sensing presents unique challenges for
209 perception: as an interactive modality, touch can
210 unintentionally shift objects during exploration,
211 and modern vision-based tactile sensors, such
212 as GelSight [22] and DIGIT [46], produce in-
213 herently high-dimensional observations. At the
214 same time, tactile sensing provides highly local-
215 ized information confined to points of contact,
216 necessitating active perception.

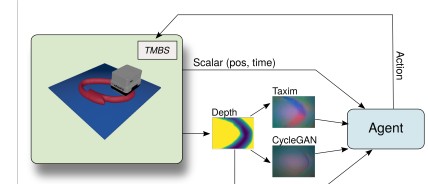

Figure 3: The agent receives the sensory information from the environment. This observation consists of scalar values and a tactile image. This can have three rendering modes, depth, Taxim, or CycleGAN.

217 The Tactile MNIST Benchmark Suite (TMBS)
218 extends `ap_gym` to tactile tasks using vision-
219 based tactile sensors. It contains four environ-
220 ments: *TactileMNIST*, where the agent must classify a 3D model of a hand-written digit (see
221 Section 3.4), *TactileMNISTVolume*, which tasks the agent to infer the volume of a given digit, *Toolbox*,
222 where the agent must estimate the pose of a tool, and *Starstruck*, in which the agent must count the
223 number of stars among other shapes (see Table 2 for an overview). In each environment, the agent
224 controls a simulated GelSight Mini tactile sensor [22] and is presented with one or more objects on a
225 platform. By interacting with the object, the agent must infer task-dependent properties, such as the

class of the object, its pose, or its volume. Particularly, aside from tactile data and proprioception, the agent does not receive any additional sensory data, so it has to infer the required property from touch alone. Although, for simplicity and performance reasons, we do not simulate the physical interaction between the sensor and the object, we shift the objects around randomly to simulate unintended object movements.

To simulate the tactile sensor, we support three rendering modes:

**Taxim:** Taxim [55] computes an approximation of the gel deformation and afterwards applies a data-driven rendering algorithm.

**CycleGAN:** With data collected on 3D printed MNIST 3D objects (see Section 3.5), we train a CycleGAN [23] for a style transfer between a depth image and tactile image [58]. The resulting images are visually much more realistic than the Taxim renderings, and thus might be beneficial for sim-to-real transfer. However, this mode is currently only available for the *TactileMNIST* environment. For more details, refer to Appendix C.

**Depth:** Here, the agent receives a depth image clipped to the GelSight gel thickness (4.25mm).

A comparison of all rendering modes is shown in Fig. 3, a visualization of the TactileMNIST digit classification task is provided in Fig. 2, and an overview of all tasks in TMBS is given in Table 2.

## 3.4 The MNIST 3D Dataset

*MNIST 3D*[3] is a collection of 13,580 auto-generated 3d-printable meshes derived from a $500 \times 500$ - pixel high-resolution MNIST variant [59] and scaled to fit in a 10x10cm square. The MNIST 3D dataset poses an exciting tactile classification challenge, as it has significant variability in shape and size within the classes, while also being large enough to facilitate learning from data. A single touch is rarely enough to classify objects from this dataset, as segments of hand-written digits are usually ambiguous. Hence, even after finding the object, the agent has to apply some strategy (e.g., contour following) to gather enough information for a successful classification. In addition to tactile sensing, this dataset could also be used as a benchmark for 3D mesh classification methods. More details on the generation of the MNIST 3D dataset can be found in Appendix D.

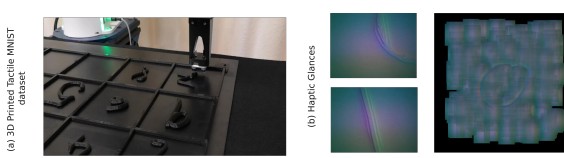

Figure 4: Data collection for the *Tactile MNIST Real Static* dataset. We mounted a GelSight Mini sensor on a Franka Research 3 (a) and collected 153,600 touches across 600 3D-printed MNIST digits. Examples of individual touches are visible in (b), and a collection of 256 touches overlayed based on their positions in (c).

## 3.5 A Large Dataset of Real Tactile Interactions

To complement simulated renders, TMBS includes a real–world, static tactile dataset captured with a GelSight sensor on 3D-printed MNIST 3D digits: the *Real Tactile MNIST Dataset*[4]. The dataset contains video sequences of 153,600 touches across 600 digits, which amounts to 256 touches per object collected in sequence. For data acquisition, we laid each 3D-printed MNIST digit in a 12x12cm grid on a rubber mat and used a Franka Research 3 robot arm [60], with a GelSight tactile sensor to press the sensor down at random locations in the cell. Once we measured a normal force exceeding 5N, we stopped pressing and registered the time stamp. To prevent degradation of the elastomer gel, we replaced the GelSight sensor's gel pad after every 76,800 touches (i.e., halfway through each dataset). Finally, we partitioned each dataset into training (90 %) and test (10 %) splits, ensuring uniform class distributions across each split. Note that we also provide two processed versions of this dataset, where we replaced the videos with still images at the time of contact, one in full resolution at 320x240px[5] and one scaled to 64x64px[6] for faster loading and training.

---

[3] https://huggingface.co/datasets/TimSchneider42/tactile-mnist-mnist3d

[4] https://huggingface.co/datasets/TimSchneider42/tactile-mnist-touch-real-seq-t256-320x240

[5] https://huggingface.co/datasets/TimSchneider42/tactile-mnist-touch-real-single-t256-320x240

[6] https://huggingface.co/datasets/TimSchneider42/tactile-mnist-touch-real-single-t256-64x64

We note that an additional challenge introduced during data collection is the possibility of the digit shifting slightly due to contact with the sensor. This variability makes the dataset more representative of real-world scenarios and provides an opportunity for methods to learn robustness to object movement and misalignment. Thus, the Real Tactile MNIST Dataset serves several key roles in the TMBS benchmark. First, it enables training of a CycleGAN for realistic simulation of tactile images and sim-to-real transfer. Second, it can be used for both pretraining and fine-tuning of learning-based perception models, enabling models to acquire basic tactile features before being deployed on a robot for active learning tasks and again facilitating sim-to-real transfer. Finally, the dataset provides a reproducible, offline benchmark for validating and comparing active perception algorithms under realistic sensor noise and material artifacts. For additional details on the data collection procedure, refer to Appendix E.

## 3.6 Evaluation Protocols

In `ap_gym` environments, there are two levels of exploration: (1) during an episode, the agent must explore to gather information, and (2), over the course of the training, the agent must explore the effects of its actions to optimize its model and policy. To disambiguate the measures of performance in these two levels, we will call the first one *exploration efficiency* and the second *sample efficiency*. Importantly, the former is a quality measure of the policy, while the latter is a quality measure of the learning algorithm. Here we borrow the term *sample efficiency* from the RL literature, where it refers to the number of environment interactions the agent needs in order to learn to solve the given task. By *exploration efficiency*, on the other hand, we refer to the efficiency with which the agent collects information within an episode.

Regarding *exploration efficiency*, we consider two measures: the *average* prediction accuracy and the *final* prediction accuracy. Here, *average* prediction accuracy means the prediction accuracy the agent exhibited throughout an episode on average, while *final* prediction accuracy means the accuracy the agent exhibited at the final step of the episode. Normally, the agent starts each episode with little to no information and then keeps gathering information as the episode progresses. Hence, for a rational agent, we expect the prediction accuracy to increase over the course of the episode and to be highest at the end of the episode. Thus, the *average* prediction accuracy could be seen as a measure of how quickly the agent explored, while the *final* prediction accuracy could be seen as a measure of how thoroughly the agent explored throughout the episode. In `ap_gym` environments, both of these measures are tracked for a number of environment-specific prediction accuracy metrics, such as classification accuracy, mean-squared-error, pose error, and others. For a detailed list of metrics for each environment, refer to Appendix F.

Approaches evaluating on `ap_gym` or the Tactile MNIST benchmark suite should report both *average* and *final* metric values over the course of the training. If applicable, the metrics should be computed on the test variants of the environments, which use the test split instead of the training split. The objective is to maximize both *sample efficiency* and *exploration efficiency*. Section 4 serves as an example for an evaluation report on `ap_gym` and Tactile MNIST environments.

## 4 Experiments

In this section, we highlight experiments across selected environments from `ap_gym` and TMBS for various baseline methods, including `TAP` [52] and `HAM` [8]. Both `TAP` and `HAM` are RL-based active perception methods and, thus, well suited for evaluation on TMBS. The main difference between them is that `TAP` employs an actor-critic RL approach in combination with a transformer architecture, while `HAM` relies on a REINFORCE gradient in combination with an LSTM model. `TAP` provides two variants: `TAP-SAC` and `TAP-CrossQ`, based on `SAC` [61] and `CrossQ` [62]. We additionally evaluate a baseline that uses `TAP`'s transformer model with PPO [63], which we call `TAP-PPO`.

We highlight experiments on four environments in total. In the CircleSquare environment, the agent can move a glimpse around an image and has to find and classify an object that can be either a circle or a square. The TactileMNIST environment tasks the agent to classify MNIST 3D models by touch alone, and in the Starstruck environment, the agent must count the number of stars (1-3) among other objects on the platform. In the Toolbox environment, the agent must find a tool and determine its 2D pose and orientation on the platform. These environments represent a diverse set of challenges, and solving them requires the agent to adopt efficient exploration strategies, which is made evident by the

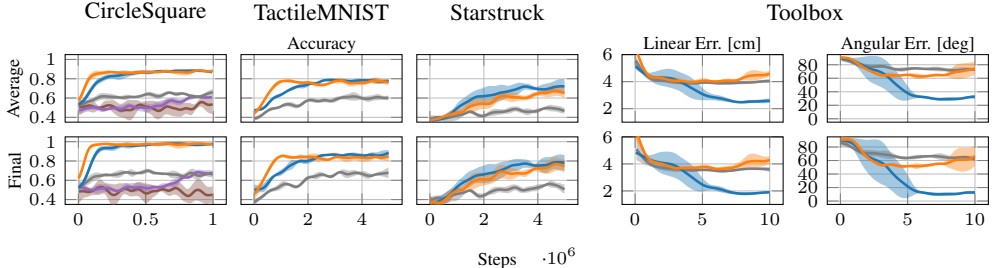

Figure 5: Average and final prediction accuracies for the baseline methods `TAP-SAC`, `TAP-CrossQ`, `TAP-PPO`, `HAM` [8], and a random baseline `TAP-RND` for the CircleSquare (`ap_gym`), TactileMNIST (TMBS), Starstruck (TMBS), and Toolbox (TMBS) environments. All methods were trained on 5 seeds for up to 10M. Shaded areas represent one standard deviation. Metrics are computed on evaluation tasks with unseen objects, except for Circle-Square and Toolbox, which have only two and one, respectively. For Starstruck, a correct classification requires predicting the exact number of stars. For Toolbox, we compute the linear and angular displacement between the prediction and the actual object pose as a metric. As `HAM` does not have a vision encoder, we evaluate it on non-tactile environments only. For `TAP-PPO`, we found it to be unstable in combination with a vision encoder, so we resort to only evaluating it on non-tactile environments as well.

consistently poor performance of `TAP`'s random baseline `TAP-RND` in Fig. 5. Further experiments and details for the training can be found in Appendix G.

As visible in Fig. 5 and Appendix G, `TAP`'s off-policy methods perform consistently better than the on-policy baselines `HAM` and `TAP-PPO`. This gap is likely due to on-policy methods generally being more sample-efficient than off-policy methods, as on-policy methods cannot reuse previously collected samples. However, despite the better performance of `TAP`, neither the `ap_gym` tasks nor the TMBS can be considered solved. `TAP` requires millions of environment interactions to learn viable exploration policies and falls short of perfect accuracy. More research in the area of sample-efficient RL and active perception is needed to improve sample efficiency to allow for the deployment of such methods in the real world.

## 5  Limitations and Conclusion

In this paper, we have introduced *Active Perception Gym* (`ap_gym`), a Gymnasium-compatible benchmark suite tailored for evaluating active perception tasks, and the *Tactile MNIST Benchmark Suite* (TMBS), which contains four tactile-specific tasks designed for robust exploration. To support these benchmarks, we have released a dataset comprising 13,580 high-resolution 3D digit models and an extensive real-world dataset of 153,600 tactile samples collected from 600 3D-printed digits using a GelSight Mini sensor. Provided as an open-source framework, `ap_gym` and TMBS offer a structured, standardized, and reproducible benchmark intended to facilitate advancements in active perception research, including efficient exploration strategies, sim-to-real adaptation through CycleGAN training, and the pretraining and fine-tuning of tactile models.

However, our benchmark has limitations, most notably the absence of online, real-world evaluation scenarios and metrics beyond the static dataset. While there are established datasets such as YCB [64] enable benchmarking of contact-rich manipulation tasks using real-world objects, our suite deliberately focuses on controlled, simulation-based exploration to ensure reproducibility and interoperability. In future work, we aim to address this limitation by designing carefully structured real-world tactile exploration experiments that extend the benchmark's relevance to physical robotic systems and support the study of sim-to-real transfer in active perception. Another key limitation is the absence of a physics engine in our simulation environment, which currently prevents modeling of more complex, contact-rich interactions. As a result, tasks such as grasping, 3D object reconstruction (where objects may tumble upon contact), object retrieval in cluttered scenes, and in-hand pose estimation remain out of scope. Extending our framework to incorporate physics engines would open the door to these richer interaction scenarios and significantly broaden the benchmark's applicability. Ultimately, we view this benchmark as a foundational resource for advancing active tactile perception and enabling the development of more robust, efficient algorithms for robotic tactile manipulation.

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
