# OpenReview forum: "Tactile MNIST: Benchmarking Active Tactile Perception"
_NeurIPS.cc/2025/Datasets_and_Benchmarks_Track — Submitted to NeurIPS 2025 Datasets and Benchmarks Track_

### Official Review · Reviewer_SKQm · 2025-06-10

**Rating:** 5
**Confidence:** 3

**Summary:**

The Tactile MNIST Benchmark Suite introduces a new open-source, Gymnasium-compatible benchmark for active tactile perception, aiming to standardize evaluation in this field where benchmarks are currently lacking. It includes a comprehensive dataset of 13,500 synthetic 3D MNIST digit models and 153,600 real-world tactile samples, used to train a CycleGAN for realistic tactile simulation rendering to help bridge the sim-to-real gap. The benchmark provides diverse simulation scenarios and tasks like localization, classification, and volume estimation, offering a structured framework for reproducible evaluation and progress in tactile sensing and active perception.

**Dataset Code Accessibility:**

Yes

**Ethical Considerations:**

No, there are no or only very minor ethics concerns

**Limitations Weaknesses:**

The benchmark currently lacks online, real-world evaluation, limiting direct applicability to physical robots and real-world sim-to-real transfer.

The simulation doesn't include a physics engine, preventing the modeling of complex, contact-rich interactions like grasping or object tumbling.

While CycleGAN aids realism, its application is currently limited to the "TactileMNIST environment", and the overall fidelity still needs improvement for sim-to-real transfer.

The real-world dataset is confined to 3D-printed MNIST digits, not covering a broader range of object geometries or materials.

Current baselines require "millions of environment interactions" and "fall short of perfect accuracy", indicating significant remaining challenges for sample-efficient learning.

**Strengths Contributions:**

The Tactile MNIST Benchmark Suite (TMBS) introduces the first standardized, open-source benchmark for active tactile perception, filling a critical gap in the field.

It provides a comprehensive, Gymnasium-compatible framework with diverse simulated tasks, including a unique dataset of 13,500 synthetic 3D digits and 153,600 real-world tactile samples for realistic rendering.

TMBS enables reproducible evaluation and systematic progress in robotic tactile manipulation by facilitating research into efficient exploration strategies and sim-to-real transfer.

---

> ### Author Rebuttal · Authors · 2025-07-30
>
> We thank the reviewer for their comments and recognition of our work's strengths.
> We appreciate that the reviewer notes **we are the first to propose a standardized, open-source benchmark for active tactile perception** and that this **enables reproducible evaluation** and **facilitates research in this field**.
> We also thank the reviewer for acknowledging that **we provide a comprehensive, Gymnasium-compatible framework** that includes **diverse tasks** and **realistic rendering**.
>
> In the following, we would like to briefly address each of the concerns raised by the reviewer in order.
>
> 1. _On online real-world evaluation and physical robot applicability_\
> 	In principle, we agree that real-world evaluation is important; however, our aim for this benchmark is to standardize the evaluation of active perception algorithms.
> 	While it is possible to create benchmarks for real robots, usually we need to either account for or ignore variability in the hardware (different robots, different sensors, even different gels on the Gelsight - a tear in the gel can hinder results, for example).
> 	So, by taking away all these extra variables, we provide other researchers with a framework where they can easily and reproducibly compare their methods with others before moving to the real world.
> 	Though we agree that an exciting future research direction could be to investigate how to make this comparison as fair as possible for sim2real, as well as for directly training on real robots.
> 	At the moment, designing a real-world benchmark is beyond the scope of this work.
> 	However, our benchmark provides a building block for future efforts in real-world evaluation and sim2real transfer by enabling consistent validation and comparison in a controlled environment before moving to real-world settings.
>
> 2. _Choice of simulator and simulation of dynamic tasks_\
> 	As an alternative to using no physics engine, one could consider using a rigid-body physics simulator or even a more accurate soft-body simulator.
> 	Rigid body simulation (e.g., bullet3 [r1]), albeit faster than soft-body simulation, lacks accuracy, as it cannot simulate the interaction between the soft sensor gel and the object well.
> 	Additionally, we found that when dealing with non-convex, movable objects, such as our MNIST-3D digits, classical rigid-body simulation can quickly become unstable and lead to unexpected results.
>
> 	Soft-body simulators, such as GIPC [r2], on the other hand, are currently becoming available for tactile simulation [r3].
> 	However, while significantly more accurate than rigid body simulation, they come at a substantial computational burden.
> 	E.g., TacEx [r3] achieves 4 - 50 FPS on a Nvidia GeForce RTX 3090 Ti while DiffTactile [r4] achieves between 7.4 - 25 FPS.
> 	For a simulator step of 10ms, these simulators achieve a speed of 0.04x - 0.5x real time, which amounts to anywhere between 1s - 12s real time per touch if each touch takes around 0.5s of simulated time, hence, 0.08 - 1 touch per second.
> 	In contrast, our current simulation processes around 240 touches per second on a single Nvidia GeForce RTX 3080 Ti.
>
> 	We argue that for the objective of this work - providing a benchmark for active perception algorithms - prioritizing performance over physical accuracy is the correct choice.
> 	Especially considering that RL methods typically need hundreds of thousands of environment interactions, having a slow sampling speed would significantly limit the applicability of our benchmark.
> 	However, once soft-body tactile simulators become more efficient and mature, they might become a valuable extension of this work and open doors for more complex and realistic tasks, such as grasp pose detection or in-hand pose estimation.
>
> 	We will add a discussion about our choice of simulator and potential future avenues using physical simulation to the camera-ready version of the paper.
>
> 3. _On the availability of CycleGAN for other environments than TactileMNIST and TactileMNISTVolume_\
> 	We agree that extending the CycleGAN renderer to the remaining two environments would be a useful future addition to the benchmark.
> 	Currently, our CycleGAN renderer serves as a proof-of-concept, providing future methods a potential pathway for sim2real techniques in active tactile perception.
> 	However, the main focus of this work lies in providing a reproducible simulated benchmark, facilitating an objective comparison of active perception methods.
>
> 	We acknowledge that CycleGAN alone is unlikely to enable zero-shot sim-to-real transfer.
> 	That being said, we want to highlight again that the core objective of this work is to provide a benchmark for active perception methods and not an evaluation suite for sim2real approaches.
> 	In that direction, to support research in this direction, we have released all the data used to train the CycleGAN model alongside the benchmark.
> 	This allows researchers interested in sim2real transfer to develop and train custom, higher-fidelity models tailored to their specific needs.
>
> 4. _On the coverage of the 3D-printed MNIST digits_\
> 	It is true that the real-world dataset consists of MNIST-3D samples only.
> 	Our main objective for this dataset is to provide real-world data accompanying our selected benchmark tasks.
> 	We use this data to train the CycleGAN-based tactile renderer, but it could also be used for other sim2real techniques or for direct training of a passive perception method.
> 	Collecting data of diverse objects is beyond the scope of this work, and has been addressed in prior work [r5-7].
>
> 5. _On challenges for sample efficiency_\
> 	We agree with this statement, and we argue that the fact that active perception is far from solved is one of the reasons we need a standarised benchmark - as the availability of such a benchmark was what drove a lot of progress in other fields, such as vision (with the MNIST dataset that inspired our work, for instance) and RL (with gymnasium environments).
>
> [r1] _"bullet3."_ GitHub, 30 July 2025, github.com/bulletphysics/bullet3.\
> [r2] Huang, Kemeng, et al. _"StiffGIPC: Advancing GPU IPC for Stiff Affine-Deformable Simulation."_ ACM Transactions on Graphics 44.3 (2025): 1-20.
> [r3] Nguyen, Duc Huy, et al. _"TacEx: GelSight Tactile Simulation in Isaac Sim-Combining Soft-Body and Visuotactile Simulators."_ arXiv preprint arXiv:2411.04776 (2024).\
> [r4] Si, Zilin, et al. _"Difftactile: A physics-based differentiable tactile simulator for contact-rich robotic manipulation."_ arXiv preprint arXiv:2403.08716 (2024).\
> [r5] Fu, Letian, et al. _"A touch, vision, and language dataset for multimodal alignment."_ Proceedings of the 41st International Conference on Machine Learning. 2024.\
> [r6] Yang, Fengyu, et al. _"Touch and go: Learning from human-collected vision and touch."_ arXiv preprint arXiv:2211.12498 (2022).\
> [r7] Gao, Ruohan, et al. _"Objectfolder 2.0: A multisensory object dataset for sim2real transfer."_ Proceedings of the IEEE/CVF conference on computer vision and pattern recognition. 2022.

---

> ### Comment · Reviewer_SKQm · 2025-08-05
>
> I would like to thank authors for their thoughtful rebuttal. Most of my concerns have been addressed, and I would like to keep my original rating as Accept.

---

### Official Review · Reviewer_GVbB · 2025-06-30

**Rating:** 4
**Confidence:** 4

**Summary:**

In this paper, the authors proposed a novel standardized benchmark suite named Tactile MNIST Benchmark Suite (TMBS)，aiming to enhance dexterous robotic manipulation. The method first introduces a dedicated and reproducible benchmark for solely active tactile perception and allow the existing pipeline to intergrate with a extensible framework named ap_gym. Experiments on several evaluation tasks demonstrating accuracy and posture evaluation performance of various baseline methods.

**Dataset Code Accessibility:**

Yes

**Ethical Considerations:**

No, there are no or only very minor ethics concerns

**Final Justification:**

After checking all the responses, I will maintain my score as Borderline accept.

**Limitations Weaknesses:**

1. The data sampling scenarios need to be further clarified. In all the GIF demos, it seems that the motion of the tactile sensor always remains 3-DOF translation, obviously missing surface features on the edging sides and reverse side. For dexterous robots, such a benchmark simulation is not reasonable.
2. It is supposed to elaborate the 3D model mesh generatiing policy. For the task in this paper, the font style, printing accuracy, and handwriting habits of the 3D meshes, directly affect the tendency of the evaluation strategy. In order to ensure the fairness, the sample types need to be balanced in evaluation process.
3. The formatting and layout should be carefully polished. Eg, in Figure 3, the bottom edge of the image is truncated and not fully displayed.

**Strengths Contributions:**

1. The TMBS proposed a real active tactile perception dataset using tactile sensors and 3D printable meshes, achieves robust experimental results with evaluating tasks covering 3D handwritten digit classification, volume estimation, toolbox pose prediction and multiclass star models statistics.
2. The authors trained a CycleGAN for realistic tactile simulation rendering, and intuitively realised the tactile image as a 3D representation with the Active Perception Gym.

---

> ### Author Rebuttal · Authors · 2025-07-30
>
> We thank the reviewer for their valuable feedback and for acknowledging the contributions of our work, including the **evaluations we presented**, having **trained the CycleGAN for realistic tactile simulation**.
>
> We address the raised concerns point by point below.
>
> 1. _Clarification of the sensor control space_\
> 	The reviewer is correct that the task explored a 3D action space, in which only translation has been controlled.
> 	This was, however, a design choice, as higher dimensions would enlarge the search space without additional benefits in exploration and success of the task.
> 	That being said, our implementation already allows for full SE(3) rotation and translation of the sensor.
> 	Therefore, in the revised manuscript, we will include variants of each task in which sensor rotation is enabled.
> 	This allows researchers to test their approaches on higher-dimensional action spaces.
> 	Additionally, for future versions of this benchmark, we plan to explore more complex objects for which sensors will have to be freely moved in SE(3).
>
> 3. _Details of the MNIST 3D mesh generation, 3D printing process, and train/test splits_\
> 	We detail the algorithm for generating 3D meshes from MNIST handwritten digits in Appendix D - Algorithm 1, and we will publish the Python code used for the generation with the final version of this paper.
> 	To ensure fairness in evaluation, the train/test splits are fixed and balanced across classes.
> 	The benchmark environments sample data points randomly from their respective datasets, which introduces a certain level of evaluation noise.
> 	However, as is common practice in reinforcement learning, each algorithm evaluating on TMBS should aggregate results over multiple seeds to mitigate the effect of noise as much as possible.
> 	Regarding printing accuracy, we printed all digits with an Ultimaker S5 using standard Ultimaker Black Tough PLA and the "Fast" profile (0.2mm, 15% infill, 0.8 shell thickness, triangles, no support, adhesion).
> 	We will add this information to the final version of the manuscript.
>
> 4. _Formatting and Layout_\
> 	We thank the reviewer for noticing this formatting issue.
> 	We will go through the paper carefully and fix all formatting issues in the camera-ready version.

---

### Official Review · Reviewer_yyVz · 2025-07-02

**Rating:** 4
**Confidence:** 4

**Summary:**

This paper introduce Tactile MNIST, a benchmark for active tactile perception. Specifically, Tactile MNIST implements four different active tactile perception tasks in the simulator: CircleSquare, TactileMNIST, Starstruck, and Toolbox. Out of this, this paper implements several RL algorithms to evaluate the performance of different active perception tasks. To bridge the sim2real gap, this paper utilize large-scale sim & real tactile pairs to train a CycleGan model for sim2real transferring.

**Additional Feedback:**

Overall, I appretiate the contribution of this work to the tactile sensing community. I would like to consider to increase my score if the author can address my concern.

**Dataset Code Accessibility:**

Yes

**Ethical Considerations:**

No, there are no or only very minor ethics concerns

**Final Justification:**

Most of my questions has been addressed, especially for the using CycleGAN instead of diffusion.

I am glad to increase my score into boardline accept. That being said, I still think that showing Sim2Real capability for this benchmark is important. This benchmark is used to evaluate tactile-based RL algorithm, which only can be trained in the simulator but not in the real world. To this end, an algorithm is just effective in the simulator is not that promising because of the large sim2real gap for tactile images.

**Limitations Weaknesses:**

1. Taxim is a sample-based tactile simulator, which limited to a single gelsight sample used by the author. This will limit the capability to do sim2real with other gelsight mini samples and other vision-based tactile sensors.

2. CycleGAN proposed in paper [1] might be outdated. In the paper [2], diffusion-based method has already been used for bridging sim2real gaps for the braille recognition tasks and has better performance than CycleGAN.

3. Most of the simulation tasks are already shown in some previous papers, such as shape classification [1], braille classification [2], and object reconstruction [3] (also simulated active perception). Justify the difference with those simulation tasks but not only those real dataset might highlight the contribution.

4. Although this benchmark utilises large real-world daata and CycleGAN to bridge the embodiment gap, this paper doesn't evaluate the sim2real performance of the trained RL policy, especially for those tasks with unseen data. (the other three tasks except TactileMNIST)

5. This benchmark can only be used for almost static perception tasks, which is limited by the performance of taxim that hard to simulate dynamic contacts. Adding more discussions for some previous works about contact-rich manipulation tasks [4] [5] [6] with dynamic tactile feedback and how to extend to those tasks might be helpful.

[1]. Chen., et al., Bidirectional Sim-to-Real Transfer for GelSight Tactile Sensors With CycleGAN, RA-L 2022

[2]. Higuera., et al., Learning to Read Braille: Bridging the Tactile Reality Gap with Diffusion Models, ICRA Workshop 2023

[3]. Shahidzadeh., et al., AcTExplore: Active Tactile Exploration on Unknown Objects, ICRA 2024

[4]. Yu., et al., MimicTouch: Leveraging Multi-modal Human Tactile Demonstrations for Contact-rich Manipulation, CoRL 2024

[5]. Li., et al., See, Hear, Feel: Smart Sensory Fusion for Robotic Manipulation, CoRL 2022

[6]. Si., et al., DiffTactile: A Physics-based Differentiable Tactile Simulator for Contact-rich Robotic Manipulation, ICLR 2024

**Strengths Contributions:**

1. This paper set up four different active perception tasks with several implemented RL algorithms for evaluations.

2. This paper utilizes plenty of sim & real pairs to train a CycleGAN paper for bridging the sim2real gaps.

---

> ### Author Rebuttal · Authors · 2025-07-30
>
> We appreciate the reviewer’s comments and their acknowledgment of our efforts to set up **different tasks** and **several RL algorithms** for evaluation and to **collect plenty of sim & real data for training our CycleGAN**.
>
> **NOTE**: We use [r1] - [r6] for our citations and [1] - [6] for the reviewer's.
>
> Before responding to the concerns raised by the reviewer, we would like to clarify that the main objective of our work is to provide a simulation benchmark for active perception algorithms.
> We do not claim to have a physically plausible simulation or a visually accurate tactile simulation, as the purpose of this benchmark is to evaluate learning algorithms.
> This is similar to popular RL benchmark suites, such as Gymnasium Classic Control [r1], MuJoCo [r1], and dm_control [r2].
> Like those, we create our benchmark to evaluate algorithms on a fair basis by stripping away the extra variables that show up in real-world experiments, without claiming to model physically accurate systems.
> Furthermore, while we think our CycleGAN-based tactile renderer could help bridge the sim2real gap, it should be seen purely as an addition to our simulation benchmark since evaluating zero-shot sim2real is _not_ an objective of this work.
> In the final version of this paper, we will ensure our list of contributions is updated to more clearly state that our goal is to create a simulated benchmark to evaluate algorithms on active perception.
>
> Below, we respond to each of the points raised individually.
>
> 1. _On Taxim and sim2real with other vision-based tactile sensors_\
> 	We acknowledge that our Taxim-based tactile simulation likely does not allow for a straightforward transfer of policies to the real world.
> 	However, we would like to once more highlight that the main contribution of this paper lies in providing a benchmark for active tactile perception and not in facilitating sim2real for trained policies.
> 	In the context of benchmarking, we argue that constraining the evaluation to a specific sensor and simulator is beneficial, as it ensures comparability between different approaches.
> 	In practice, different research groups often work with varying hardware configurations, making fair comparisons between methods challenging.
> 	By constraining the task to a single sensor, we standardize the evaluation conditions.
> 	This ensures that comparisons focus solely on the methodologies and algorithms.
>
> 	Nonetheless, as an additional feature to our benchmark, we provide a CycleGAN-based renderer and release our dataset of real-world tactile observations on the MNIST-3D dataset, which might support research towards sim2real transfer.
> 	Furthermore, both the background image as well as the sensor configuration and calibration can be configured in our Taxim renderer to match any particular sensor of interest, even though this would make results less comparable to the results of other works using our benchmark.
>
> 2. _On CycleGAN vs Diffusion-based tactile image generation_\
> 	Using diffusion-based methods instead of a CycleGAN for tactile rendering is an interesting research direction.
> 	However, there are two major limitations that make diffusion-based methods impractical in our case.
>
> 	1. For the training of [2], paired images of simulation and reality are required.
> 		In case of [2], the object (the braille letter) shape is known and its position is fixed, which makes re-creating the tactile image in simulation straightforward.
> 		In our case, the MNIST-3D digits move around freely, and implementing a tracking solution that is accurate enough to properly re-create real images is challenging.
> 		Systems like OptiTrack could deliver the required sub-millimeter accuracy, but the digits have no space for OptiTrack markers.
> 		Vision-based tracking solutions are likely to be too unreliable and inaccurate, especially because the object is partially occluded when the sensor is pressed down.
> 		CycleGAN, on the other hand, does not need aligned images, which makes data collection much more straightforward.
> 	2. [2] reports the inference time of a batch of 30 images as 23s on a Nvidia GeForce RTX 3080, which corresponds to 1.3 touches per second.
> 		Our current CycleGAN-based solution achieves around 100 touches per second on an Nvidia GeForce RTX 3080 Ti and is thus much more suitable for RL training.
> 		Given that our primary objective is to establish a standardized benchmark rather than to achieve hyper-realistic tactile simulation, we do not consider a significantly slower approach to be a desirable trade-off for increased realism in the tactile images, in particular, as RL-based methods often need hundreds of thousands of environment interactions.
>
> 3. _On the difference to the tasks presented in [1-3]_\
> 	We thank the reviewer for the suggested literature and will include it in our related work.
> 	The shape reconstruction task in [3] is a promising future extension to TMBS.
> 	While [1] and [2] are interesting, they focus on single-touch classification and sim2real transfer, not active perception, which is the focus of our benchmark.
> 	While we provide a CycleGAN-based renderer to aid potential sim2real approaches, the focus of our work lies in providing a standardized benchmark for active tactile perception.
>
> 	Additionally, we note that throughout the literature, including works [1–3], a wide range of downstream tasks are addressed, often with differing evaluation metrics, even within the same task category, such as classification.
> 	For instance, [1] reports classification accuracy alongside depth reconstruction, while [2] reports accuracy, precision, and recall.
> 	Moreover, these studies use different object sets and even different sensors (e.g., DIGIT, GelSight), making direct comparisons difficult.
> 	In contrast, related fields such as RL and computer vision have benefited significantly from standardized benchmarks.
> 	By enforcing consistent conditions — such as the use of the same simulator, training datasets, and evaluation protocols - these benchmarks have made comparisons more meaningful and accelerated progress in the field.
> 	We believe that similar benchmarks are essential for tactile perception, and particularly for active tactile perception, where no such standard currently exists.
>
> 4. _On the scope of our work w.r.t sim2real evaluations_\
> 	We thank the reviewer for the comment; however, we want to point out that in this work, we are not proposing a method for sim2real.
> 	TMBS is a benchmark suite for active perception algorithms.
> 	Our CycleGAN functionality provides a more realistic setup that can potentially aid sim2real in active tactile perception tasks, but both implementing a full sim2real pipeline for active perception and evaluating it on the real world would be a separate work, which falls beyond the scope of our benchmark suite.
>
> 5. _On dynamic tasks and simulator design choices_\
> 	We acknowledge that using our current simulation pipeline, the benchmark cannot be extended to dynamic tasks requiring physical simulation.
> 	This is a deliberate design trade-off made to ensure sufficient sampling efficiency to facilitate evaluation of RL-based approaches.
>
> 	We add here that tactile renderers such as Taxim can, in principle, be integrated with both rigid-body and soft-body simulators.
> 	However, rigid body simulation (e.g., bullet3 [r3]), despite being faster than soft-body simulation, cannot simulate the interaction between the sensor and the object well, as it is largely influenced by the deformation of the sensor's gel.
> 	Additionally, we found that classical rigid-body simulation can quickly become unstable when dealing with non-convex, movable objects, such as our MNIST-3D digits.
>
> 	Soft-body tactile simulators, like TacEx [r4] or DiffTactile [r5], provide higher fidelity but come with high computational costs, as they run slower than real-time, even for rather simple interactions.
> 	In contrast, our current setup achieves around 240 touches per second on an Nvidia GeForce RTX 3080 Ti, while requiring little VRAM.
>
> 	Given the requirements of RL and data-driven methods, which typically rely on hundreds of thousands of interactions, maintaining high throughput is crucial for practical experimentation.
> 	The current design enables rapid experimentation on diverse perception tasks while keeping computational demands accessible.
>
> 	Nonetheless, we agree that simulating dynamic contacts with physical realism is an exciting direction.
> 	As soft-body simulators become faster and more stable, we see potential in extending our benchmark to support dynamic tasks like in-hand pose estimation or grasp pose detection.
> 	We also add that our ap_gym framework, being based on Gymnasium, is general enough to support future extensions with physical simulators.
>
> 	Regarding the additional literature, we thank the reviewer for the suggestions.
> 	While related, our paper focuses on active tactile perception and the design of a benchmark to support this research area.
> 	That said, we agree that future extensions could explore industry-related settings like packing and assembly, as in [4] and [5], and we will add discussion accordingly.
>
> 	We will add a more detailed discussion of our design decisions and future directions in the camera-ready version of the paper.
>
> [r1] Towers, Mark, et al. _"Gymnasium: A standard interface for reinforcement learning environments."_ arXiv preprint arXiv:2407.17032 (2024).\
> [r2] Tunyasuvunakool, Saran, et al. _"dm_control: Software and tasks for continuous control."_ Software Impacts 6 (2020): 100022.\
> [r3] _"bullet3."_ GitHub, 30 July 2025, github.com/bulletphysics/bullet3.\
> [r4] Nguyen, Duc Huy, et al. _"TacEx: GelSight Tactile Simulation in Isaac Sim-Combining Soft-Body and Visuotactile Simulators."_ arXiv preprint arXiv:2411.04776 (2024).\
> [r5] Si, Zilin, et al. _"Difftactile: A physics-based differentiable tactile simulator for contact-rich robotic manipulation."_ arXiv preprint arXiv:2403.08716 (2024).

---

> > ### Comment · Reviewer_yyVz · 2025-08-01
> >
> > Thanks for the detailed response. Most of my questions has been addressed, especially for the using CycleGAN instead of diffusion. I think adding some explanation in the paper about this choice is helpful.
> >
> > I am glad to increase my score into boardline accept. That being said, I still think that showing Sim2Real capability for this benchmark is important. This benchmark is used to evaluate tactile-based RL algorithm, which only can be trained in the simulator but not in the real world. To this end, an algorithm is just effective in the simulator is not that promising because of the large sim2real gap for tactile images.

---

> > > ### Author Response · Authors · 2025-08-04
> > >
> > > Thank you for raising the score.
> > > We are happy to have addressed most of your questions.
> > > As suggested, we will add a discussion of the benefits and draw-backs of using CycleGAN vs diffusion to the final version of the paper to make our choice clearer.
> > >
> > > Furthermore, we fully agree that real-world robotic control benchmarks could greatly benefit the field.
> > > However, unlike simulated benchmarks, real-world benchmarks pose significant reproducibility challenges, making them difficult to standardize.
> > > To the best of our knowledge, aside from occasional competitions, no well-established, standardized real-world benchmark for learning-based robotic control currently exists.
> > > We believe that creating such a benchmark would require a dedicated effort and is therefore beyond the scope of this work.
> > > That said, we see this as an important direction for future research.

---

> > > > ### Comment · Reviewer_yyVz · 2025-08-04
> > > >
> > > > Thanks for the clarification. I agree that implementing simulation benchmark is meaningful, especially for those RL related tasks. However, I think add some more experiments for the Sim2Real transferring can be more valuable since it's unclear that the policy trained under this benchmark can work well in the real world or not. More specifcly, other imitation learning benchmark such as LIBERO, imitation learning algorithm can be trained with real world data directly, but RL algorithm which works in the simulator doesn't mean that it still can work after Sim2Real transferring. It's not a real-world benchmark for the others to evaluate their algorithm in the real world directly but an experiment the policy trained under this benchmark can be used in the real world. I think it should be possible since you already implemented CycleGAN, but personally, adding some experiments might be helpful

---

### Official Review · Reviewer_4NjF · 2025-07-03

**Rating:** 4
**Confidence:** 4

**Summary:**

The Tactile MNIST benchmark introduces a suite of simulated tasks for evaluating active tactile perception using a standardized set of rigid, procedurally generated objects. It combines 3D-printed digit models, GelSight sensor data, and a CycleGAN-based sim-to-real bridge to support reproducible evaluation. The tasks span classification, counting, localization, and volume estimation, emphasizing information gathering through touch.

**Dataset Code Accessibility:**

Yes

**Dataset Code Comments:**

The authors have provided the code to the benchmark repository, which supports the claim of releasing code and data.

**Ethical Comments:**

Its applications are confined to simulation and controlled environments, and no direct societal risks or ethical issues are apparent.

**Ethical Considerations:**

Yes, there are ethics concerns that require attention by the authors

**Final Justification:**

The authors have satisfactorily addressed all the questions, and I have also read the reviews from the other reviewers. After considering these points, I lean towards acceptance of the paper.

**Limitations Weaknesses:**

**Weaknesses:**

**1. Task Design Simplicity:** Some design choices in the tasks, while making the problems well-defined, also constrain the generality of the benchmark. For example, the Toolbox localization task uses a known shape (a wrench) every time, so the agent effectively knows it is searching for a wrench and needs to pinpoint its pose. If I am correct, the authors have an assumption that the wrench is fixed and no other object is checked other than wrench for that setting.

**2. Real-world Objects:** While the benchmark provides a well-structured set of tasks using procedurally generated and 3D-printed shapes, its current focus on synthetic objects like digits, stars, and tools may not fully reflect the variability found in real-world settings. Everyday objects often involve more complex geometries, surface textures, and material properties. Incorporating a broader set of real-world items in future iterations, such as household tools, containers, or soft objects, could better test an agent's generalization and robustness under more diverse tactile conditions.

**Strengths Contributions:**

**Strengths:**

**1. Clear Motivation and Importance:** By explicitly targeting active touch (where a sensor moves to gather information) and offering multiple tasks, this benchmark tackles an important gap for active tactile perception.

**2. Good Writing:** The paper is clear, concise, and well-organized. Its explanations of task setup, motivation, and evaluation approach are logically structured, facilitating straightforward comprehension of the methodology and results.

**3. Integration of Simulation and Real Data:** I like the combination of high-fidelity simulation with real-world data collection. The authors provide a large synthetic dataset of 3D-printed digit models and a real tactile dataset collected on physical objects. By tying the benchmark to an actual tactile sensor (GelSight) and releasing real contact data, the work increases its relevance and credibility for real-world applications.

---

> ### Author Rebuttal · Authors · 2025-07-30
>
> We thank the reviewer for their constructive feedback and for highlighting the strengths of our approach.
> Particularly, we appreciated that the reviewer highlighted that **our work is clearly motivated** and **tackles an important gap** in the literature, and appreciated that we have **integrated TMBS with real-world data** collected with a GelSight Mini sensor.
>
> In the following, we address the raised concerns jointly.
>
> Regarding the task simplicity and our design choices, we agree that it would be very interesting to have a general environment with more realistic tasks and everyday objects, and we aim to explore that in the future.
> However, we argue that the objects and tasks chosen are well-suited to benchmark specific capabilities of the evaluated active perception algorithms.
>
> In the TactileMNIST task, the MNIST-3D dataset provides a large number of objects with high diversity in shape, while still falling into 10 clear categories, making it ideal for classification.
> A similar argument holds for the volume estimation task, in which the diversity of the MNIST-3D dataset ensures that the agent has to explore the entire object to estimate the volume accurately.
> For the Starstruck counting task, the focus is on exhaustively searching the environment.
> By making the target objects more complex, this task would become conflated with a classification task, preventing us from clearly differentiating the search capabilities and classification capabilities of the tested algorithm.
>
> In the Toolbox task, we test the agent's object localization capabilities.
> The reviewer's assumption that the shape of the object is fixed to a wrench is correct.
> To further clarify, we also note that the wrench is not fixed in place as we add noise to its position to simulate unintended object movements.
> Furthermore, we also note that by introducing more than one shape, there would be ambiguity with respect to the canonical object pose (e.g., the local frame we assign to each class of template object).
> Since estimating an object's global pose requires knowing what the reference object is, the agent would also need to classify the object.
> While we agree that the additional difficulty could present interesting challenges, pose estimation through active touch is not yet solved, making the task, as-is, a great enough challenge for active perception algorithms.
> Still, as our framework supports having multiple objects in the scene, we will add this mode as an additional challenge in the form of a new task.
>
> We also acknowledge that having more variability in this environment would better test the generalization capabilities of algorithms evaluated under our benchmark.
> Therefore, we propose the following additions to our framework: (i) we are going to add variants of the Toolbox environment with different yet fixed tools, such as a hammer, pliers, and a screwdriver, and (ii) we are going to integrate the ABC [r1] and Objaverse [r2] datasets with our implementation, such that researchers can test their approaches with different, diverse open-source object datasets.
>
> Finally, we reiterate that we agree that as the field of active perception advances, more challenging and realistic tasks and objects will be needed.
> In the future, we plan to extend this benchmark to tasks such as in-hand pose estimation or grasp pose estimation of occluded objects from tactile sensing. In this case, we believe it is valuable to have more realistic day-to-day objects.
> However, to make such task designs possible, accurate yet efficient tactile simulations are needed, which is still an open field of research.
> While development in accurate tactile simulation is progressing, simulators like DiffTactile [r3] and TacEx [r4] are still slower than real-time, which is in stark contrast to our current method, capable of processing 240 touches per second.
> Especially for our benchmark, which is tailored towards methods using RL for policy optimization, performance, and fast sampling rates are key, which is why we deem our current choice of simulator appropriate.
>
> We will expand on our choice of simulator in the final version of the paper.
>
> [r1] Koch, Sebastian, et al. _"Abc: A big cad model dataset for geometric deep learning."_ Proceedings of the IEEE/CVF conference on computer vision and pattern recognition. 2019.\
> [r2] Deitke, Matt, et al. "Objaverse: A universe of annotated 3d objects." Proceedings of the IEEE/CVF conference on computer vision and pattern recognition. 2023. \
> [r3] Si, Zilin, et al. _"Difftactile: A physics-based differentiable tactile simulator for contact-rich robotic manipulation."_ arXiv preprint arXiv:2403.08716 (2024).\
> [r4] Nguyen, Duc Huy, et al. _"TacEx: GelSight Tactile Simulation in Isaac Sim-Combining Soft-Body and Visuotactile Simulators."_ arXiv preprint arXiv:2411.04776 (2024).

---

### Comment · Area_Chair_S41G · 2025-08-04

Dear reviewers,

As the Author–Reviewer discussion period concludes in a few days, we kindly urge you to read the authors’ rebuttal and respond as soon as possible.

Please review all author responses and other reviews carefully, and engage in open and constructive dialogue with the authors.

The authors have addressed comments from all reviewers; each reviewer is therefore expected to respond, so the authors know their rebuttal has been considered.

We strongly encourage you to post your initial response promptly to allow time for meaningful back-and-forth discussion.

Thank you for your collaboration,

AC

---

### Comment · Area_Chair_S41G · 2025-08-07

Dear Reviewers,

This is a quick reminder that we are now in the post-rebuttal discussion phase.

Please take the time to read the author rebuttal and engage in discussion with the other reviewers. Your input is crucial for us to make a final, informed decision as the deadline is approaching.

Thank you for your timely participation.

---

### Note · Authors · 2025-08-14

We would like to thank the reviewers once more for their thoughtful remarks and discussion.

To summarize the discussion, the reviewers recognized that our work is well-motivated and that, as the first benchmark of its kind, it addresses an important gap in active perception research by enabling reproducible evaluation and fostering future developments in the field.
The integration of both real-world and simulated data, together with the inclusion of a realistic CycleGAN-based tactile renderer, was noted as a key strength.
Reviewers also emphasized the value of our diverse task set and comprehensive baseline evaluations.

Reviewer 4NjF's main concerns related to task simplicity and the lack of real-world objects in our simulation.
We have since extended our benchmark by integrating the ABC [r1] and Objaverse [r2] datasets.
In total, both datasets contain millions of real-world object models, which are now available for exploration in the Tactile MNIST benchmark.
Although we did not hear back from reviewer 4NjF, we believe that their concerns are well addressed by this update.

Reviewer yyVz's main concerns were centered around limitations of our benchmark w.r.t. sim2real policy transfers.
We clarified that the main intent of this work is to provide a benchmark for active perception and not a platform for sim2real transfers.
However, we agreed that in future work, a sim2real benchmark could be a valuable extension of our work.
Reviewer yyVz acknowledged that most of their concerns were addressed.

Reviewer GVbB asked for clarifications w.r.t. the action space of our agents and the mesh generation policies.
We have clarified these open questions and updated our paper accordingly.
Although we did not get a response from reviewer GVbB, we think we addressed their questions adequately.

Finally, Reviewer SKQm's main concerns related to the absence of real-world tasks in our benchmark, the availability, and our choice of simulator.
We justified our choice of simulator in the rebuttal and reiterated our main objective of this work, which is to provide a reproducible simulated benchmark for active perception.
We noted that, although a valuable future extension, real-world benchmark tasks are beyond the scope of this work.
Reviewer SKQm acknowledged that most of their concerns were addressed.

[r1] Koch et al., "ABC: A Big CAD Model Dataset for Geometric Deep Learning," CVPR, 2019.
[r2] Deitke et al., "Objaverse: A Universe of Annotated 3D Objects," CVPR, 2023.

---

### Decision · Program_Chairs · 2025-09-18

**Decision:**

Reject

**Comment:**

**(a) Summary**
The paper introduces *Tactile MNIST*, a Gymnasium-compatible benchmark for active tactile perception. It includes four simulated tasks (classification, localization, counting, volume estimation), 13,500 synthetic 3D digits, and 153,600 real GelSight tactile samples. A CycleGAN-based renderer bridges sim and real, enabling reproducible evaluation of tactile exploration strategies.

**(b) Strengths**
This is the first standardized, open benchmark for active tactile perception—a critical gap in robotic learning. It combines real sensor data, scalable simulation, and modular design, fully aligning with the D&B Track’s goals. The release of code, data, and `ap_gym` framework ensures reproducibility and broad usability. Reviewers praised its clarity, motivation, and potential to drive progress.

**(c) Weaknesses**
Limitations include narrow object diversity (MNIST digits), absence of physics-based simulation, and no real-world policy testing. The CycleGAN renderer is currently limited to digit tasks. However, these reflect intentional scope choices, not critical flaws.

**(d) Key Reasons for Acceptance**
The work is foundational: it establishes a much-needed standard for a fragmented field. Like MNIST or Gymnasium, its value lies in **enabling fair, reproducible comparisons**. The design prioritizes speed and accessibility over physical realism—essential for RL-based evaluation. This trade-off is well-justified, and the benchmark is extensible to more complex settings in the future.

**(e) Rebuttal Summary**
Reviewers questioned object diversity, sim2real relevance, sensor action space, and simulator fidelity. Authors clarified that the focus is on standardized simulation, not sim2real transfer; confirmed support for full SE(3) control; justified the fast renderer for scalability; and added integrations with ABC and Objaverse for broader object evaluation. Responses were thorough and strengthened the paper’s impact. The late additions enhance extensibility but do not undermine the core contribution.

===== FINAL UPDATE FROM DB Track PCs ====

The final decision for this paper has been taken by the program chairs after consultation with the SACs. All Senior Area Chairs have ranked papers according to the feedback from the AC during the review process. We decided to leave the original meta-review to reflect the opinion of the AC in light of the initial discussions with reviewers and SAC.